# The Prevalence of Cardiometabolic Health Risk Factors among Airline Pilots: A Systematic Review

**DOI:** 10.3390/ijerph19084848

**Published:** 2022-04-16

**Authors:** Daniel Wilson, Matthew Driller, Ben Johnston, Nicholas Gill

**Affiliations:** 1Te Huataki Waiora School of Health, The University of Waikato, Hamilton 3216, New Zealand; daniel.wilson@toiohomai.ac.nz (D.W.); nicholas.gill@waikato.ac.nz (N.G.); 2Faculty of Health, Education and Environment, Toi Ohomai Institute of Technology, Tauranga 3112, New Zealand; 3Sport and Exercise Science, School of Allied Health, Human Services and Sport, La Trobe University, Melbourne 3086, Australia; 4Aviation and Occupational Health Unit, Air New Zealand, Auckland 1142, New Zealand; ben.johnston@otago.ac.nz; 5New Zealand Rugby, Wellington 6011, New Zealand

**Keywords:** aviation medicine, occupational health, morbidity, noncommunicable disease risk, risk factors, modifiable risk

## Abstract

Background: The occupational demands of professional airline pilots such as shift work, work schedule irregularities, sleep disruption, fatigue, physical inactivity, and psychological stress may promote adverse outcomes to cardiometabolic health. This review investigates the prevalence of cardiometabolic health risk factors for airline pilots. Methods: An electronic search was conducted utilizing PubMed, MEDLINE (via OvidSP), CINAHL, PsycINFO, SPORTDiscus, CENTRAL, and Web of Science for publications between 1990 and February 2022. The methodological quality of included studies was assessed using two quality assessment tools for cross-sectional and clinical trial studies. The prevalence of physiological, behavioral, and psychological risk factors was reported using descriptive analysis. Results: A total of 48 studies derived from 20 different countries, reviewing a total pooled sample of 36,958 airline pilots. Compared with general population estimates, pilots had a similar prevalence for health risk factors, yet higher sleep duration, lower smoking and obesity rates, less physical activity, and a higher overall rate of body mass index >25. Conclusions: The research reported substantial prevalence >50% for overweight and obesity, insufficient physical activity, elevated fatigue, and regular alcohol intake among pilots. However, the heterogeneity in methodology and the lack of quality and quantity in the current literature limit the strength of conclusions that can be established. Enhanced monitoring and future research are essential to inform aviation health practices and policies (Systematic Review Registration: PROSPERO CRD42022308287).

## 1. Introduction

Cardiometabolic noncommunicable diseases (NCDs) such as cardiovascular disease (CVD), stroke, type 2 diabetes (T2D), and their primary risk factors are a leading public health concern that produce significant and growing economic costs globally [1]. The leading cause of mortality worldwide is CVD [2], which has been reported as the most frequent cause of permanent groundings among Korean airline pilots [3]. Cardiovascular and cerebrovascular incidents have also been reported among the most prevalent causes of flight incapacitation in the United Kingdom [4].

Airline pilots experience unique occupational demands which may promote adverse outcomes to cardiometabolic health, including shift work, work schedule irregularities, sleep disruption, fatigue, the sedentary nature of the job, and stress demands associated with flight safety [3,5,6,7,8]. Cardiometabolic diseases are associated with numerous modifiable risk factors across physiological, behavioral, and psychological domains [1]. Central and systemic obesity, hypertension, dyslipidemia, hyperglycemia, insulin resistance, and adipose dysfunction are among prevalent physical risk factors associated with increased risk of CVD and T2D [1,9]. Modifiable behavioral risk factors [10] such as unhealthy diet, physical inactivity, excessive alcohol consumption, and tobacco smoking, along with psychological risk factors including high fatigue [11] and depression [12], are each independently established as risk factors for cardiometabolic diseases.

To date, no systematic reviews have been published pertaining to the evaluation of modifiable health risk factor prevalence among airline pilots. Estimations of health risk prevalence are important for monitoring of trends and to inform risk reduction interventions; hence, the aim of the current review was to critically analyze the global literature to quantify the prevalence of modifiable cardiometabolic health risk factors among commercial airline pilots. The findings from this review may be valuable to inform aviation health practices and policies for supporting pilot health, enhancing flight operation safety, and identifying deficiencies within the literature base to inform future research.

## 2. Materials and Methods

### 2.1. Protocol

This systematic review was conducted according to the guidelines of the Preferred Reporting Items for Systematic Review and Meta-Analysis Protocols (PRISMA-P) [13]. The protocol of this systematic review was registered with the International Prospective Register of Systematic Reviews (PROSPERO, CRD42022308287).

### 2.2. Literature Search

An electronic search was conducted utilizing PubMed, MEDLINE (via OvidSP), CINAHL, PsycINFO, SPORTDiscus, CENTRAL, and Web of Science. A broad search strategy was implemented to gather literature published between 1 January 1990 and 28 February 2022. Key terms incorporated in the search string were relating to airline pilots and cardiometabolic health risk prevalence (see Table 1). The search was limited to peer-reviewed publications in English. Eligible publications were extracted, and their reference lists were manually checked for potentially relevant studies. The reference lists of existing review articles pertaining to aviation medicine were also cross-checked for relevant articles. 

### 2.3. Eligibility Criteria

Publications were identified for inclusion on the basis of population, literature type, publication date, and cardiometabolic health risk eligibility. The population criteria for inclusion were fixed-wing pilots (airline, commercial, civilian), and no restrictions were placed on fleet type (short-haul, long-haul, mixed-fleet). Articles were excluded if they included pilots with <1 year experience of being a pilot, as well as those who worked part-time, were a helicopter pilot, or worked in noncivil aviation roles (Air Force, military, army, or private), and if they were published before 1990. Literature sources that met inclusion were peer-reviewed original articles (retrospective, prospective, cross-sectional, case–control, cohort, and experimental), and other sources were excluded, e.g., literature reviews, commentaries, and editorials. To be eligible for inclusion, publications had to report on at least one of the following cardiometabolic health risk markers: blood pressure (BP), body composition (body mass, body mass index [BMI], waist circumference, waist-to-hip ratio, body fat percentage, lean mass percentage, visceral adiposity), glycemic control (fasting or postprandial glucose, HbA1c), insulin (fasting, insulin sensitivity, or insulin resistance), inflammation (C-reactive protein, inflammatory markers), blood lipid panel (total cholesterol (TC), low-density lipoprotein (LDL) cholesterol, high-density lipoprotein (HDL) cholesterol, and triglycerides (TG)), microalbuminuria, endothelial or microvascular dysfunction, alcohol consumption, smoking, dietary behaviors (fruit and vegetable intake, high-energy-dense intake, high-saturated-fat intake, high sugar intake, or low-fiber), physical activity (sedentary behavior, moderate-to-vigorous physical activity (MVPA), or daily steps), cardiorespiratory fitness (submaximal or maximal oxygen consumption [VO_2_]), sleep (hours per night, sleep quality), psychosocial stress (stress, depression, anxiety, and fatigue), and self-rated health.

To avoid including studies involving work duty-induced inflation of cardiometabolic risk prevalence, studies pertaining to outcome measures recorded preceding (<24 h), during, or acutely following (<48 h) long-haul flights were excluded. Where available, nonflight duty baseline data from these studies were utilized. Studies reporting data exclusively on pilot subpopulations (e.g., diabetic or obese pilots) were excluded. A hand search of recent issues of prominent aviation journals was conducted to screen for any recently published articles that were not yet indexed and apparent on the systematic search.

### 2.4. Screening Process

The lead author conducted the initial literature search, and results were downloaded and imported into Endnote citation software (Endnote x9, Clarivate Analytics, Philadelphia, PA, USA) for collation and duplicate removal. Thereafter, articles were exported to Microsoft Excel (Microsoft^®^ Excel version 16.54) for further removal of duplicates and subsequent eligibility screening. Initial title and abstract screening was conducted by one author and cross-checked by a second reviewer. Subsequently, potentially eligible articles from the initial screening progressed to full-text evaluation of eligibility for inclusion. Discrepancies in outcomes between the reviewers were resolved via discussion and consultation with a third reviewer.

### 2.5. Methodological Quality Assessment

The methodological quality of publications included for this review was independently assessed by two reviewers. The risk-of-bias quality assessment checklist (adapted from Hoy and colleagues [14]) was utilized for evaluation of cross-sectional studies, which consisted of four external validity items and six internal validity items. Clinical trials were evaluated utilizing the risk-of-bias tool from Cochrane [15]. The summative quality assessment for each publication was expressed as being of low quality (high risk of bias), moderate quality (high risk of bias), or high quality (low risk of bias). Consistent with the Grades of Recommendation, Assessment, Development, and Evaluation and Cochrane approaches, total scores for the cross-sectional study assessment were grouped as the following thresholds: very high risk of bias (0–4 points), high risk of bias (5–6 points), or low risk of bias (7–10 points). Clinical trials were rated as ‘high’, ‘low’, or ‘unclear’ for seven items: random sequence generation, allocation concealment, blinding of participants and personnel, blinding of outcome assessment, incomplete outcome data, selective outcome reporting, and outcome-specific evaluations of risk of bias.

### 2.6. Data Extraction

The study country, aim, design, participant characteristics, outcomes of interest, and instruments for included publications were extracted (Appendix A). If necessary, additional publication information was sought from trial registries, article supplementary materials, or direct contact with article authors. Study data were independently extracted and coded by one reviewer, and a second reviewer independently extracted and coded 20% of the included studies for process cross-evaluation. Any discrepancies between reviewers were resolved via discussion and consultation with a third reviewer if necessary. For clinical trials included in our analysis, descriptive data were extracted from their reported baseline data, and post-intervention data were not included. For between-group studies that only reported subgroup descriptive statistics (e.g., interventional and control), we computed the combined population mean using Cochran’s formula [16].

### 2.7. Analysis of Data

The prevalence of cardiometabolic health risks was reported using descriptive analysis. Available data were sought and extracted from included publications for descriptive analysis, including one or multiple of the following available statistical metrics: mean descriptive statistics, prevalence proportions, incidence rates, standardized incidence ratios, prevalence ratios, odds ratios, risk ratios, or scoring outcomes derived from relevant self-report instruments. The meta-analysis estimates for proportions and descriptive statistics for cardiometabolic health risk factors were calculated by weighing the studies according to their sample size within pooled samples. A 95% confidence interval was presented alongside pooled prevalence statistics. Meta-analyses were not conducted for some cardiometabolic risk factors due to a low number of studies reporting the parameter of interest (*n* < 4) or due to methodological heterogeneity. Data were entered into an Excel spreadsheet (Microsoft, Seattle, WA, USA) and then imported into statistical software SPSS v28 for Windows (IBM, New York, NY, USA), where meta-analysis interpretation was performed.

## 3. Results

### 3.1. Study Selection

The search strategy produced 6138 unique results, 107 of which were deemed potentially eligible at primary screening. After full-text reviews, 48 passed eligibility evaluation for inclusion. A PRISMA flowchart depicting stages of the selection process is illustrated in Figure 1.

### 3.2. Study Characteristics

The 48 studies involved a total of 36,958 participants, included in 46 cross-sectional studies and three clinical trials (Figure 2). The characteristics of the included studies are summarized in Appendix A. Across all studies, males represented 96% of participants. The mean age of participants was 40 ± 11 years according to 35/48 studies which reported the mean age. The most prevalent age range reported in the remaining studies was 35–45 years. Twenty-five studies reported self-report subjective data, 14 utilized a combination of self-report subjective and objective data, and five reported only objective data. The included studies were conducted in 20 different countries or regions, including Brazil (five), China (five), New Zealand (four), Finland (three), Indonesia (three), Sweden (three), the United Kingdom (three), the United States (three), Korea (two), the Netherlands (two), Portugal (two), and one study each from Arab states, Australia, Europe, Germany, India, Oceania, Saudi Arabia, Spain, and Thailand. Four studies involved participants from numerous countries.

### 3.3. Quality of Reviewed Articles

The results of the risk-of-bias assessment are displayed in Table 2 and Table 3. Of the 48 publications included in the review, four were considered of low methodological quality with a high risk of bias and 13 were considered of high methodological quality with a low risk of bias. Weak external validity was apparent for most cross-sectional studies, with a paucity of random sampling (*n* = 39) and high nonresponse bias (*n* = 33) as leading factors. Lacking reliability and validity of outcome measures (*n* = 17) and inappropriate observed prevalence period (*n* = 14) were prominent factors of poor internal validity among cross-sectional studies. The three clinical trials reviewed ranged from low to moderate quality, all exhibiting high risk of bias for allocation concealment and blinding of participants.

### 3.4. Physiological Cardiometabolic Risk Factors among Pilots

Twenty-eight studies investigated physiological cardiometabolic risk factors. From the 22 studies reporting BMI, 12 were objectively measured and 10 were based on self-report data. The overall objectively measured BMI (*n* = 20,279) pooled mean was 26.1 ± 3.0 kg/m^2^ and the overall subjective BMI (*n* = 3710) pooled mean was 24.7 ± 3.1 kg/m^2^. For females, one study (*n* = 661) reported an objectively measured BMI of 23.9 kg/m^2^ (20.0–27.7), and another (*n* = 32) reported a subjective BMI as 22.7 kg/m^2^.

Eleven studies investigated the prevalence of overweight and obesity; five (*n* = 19,171) were objectively measured and six (*n* = 3309) were based on self-reporting from participants. The pooled mean for objective measures of overweight and obesity were 47.5% (47.4–47.5%) and 11.6% (11.6–11.7%), respectively. One study reported obesity only, revealing a prevalence of 20% [6]. The pooled mean for subjective measures of overweight and obesity was 43.6% (43.3–43.9%) and 12.4% (11.9–12.9%), respectively. The overall pooled prevalence of overweight plus obesity was 59.1% (59.0–59.2%) for objective measures and 56.0% (55.5–56.5%) for subjective measures. The combined pooled prevalence from subjective and objective measures for overweight, obesity, and overweight plus obesity was 46.8% (46.7–46.9), 11.7% (11.6–11.8%), and 58.6% (58.5–58.7%), respectively. One study [32] (*n* = 661) reported the prevalence of overweight and obesity for females as 28% and 6%, respectively. The prevalence of metabolic syndrome was reported by two studies, ranging from 15% [21] to 38% [27]. Furthermore, these studies reported objectively measured central obesity (>102 cm) prevalence as 18% [21] and 64% [27]. Only one study investigated C-reactive protein levels, reporting a mean hs-CRP serum level of 1.68 ± 1.79 (mg/L) [21].

Four studies (*n* = 16,327) reported the prevalence of hypertension (BP ≥ 140/90 mmHg) from objective measurement as 29% [32], 28% [27], 26% [56], and 11% [23], with a pooled prevalence of 27.6% (27.5–27.7%). Furthermore, one study (*n* = 303) reported the prevalence of elevated BP (≥130/85 mmHg) as 38% [21]. Derived from four studies [3,27,56,57], the objectively measured pooled mean systolic blood pressure (SBP) was 126 ± 14 mmHg, and the objectively measured pooled mean diastolic blood pressure (DBP) was 79 ± 9 mmHg. The prevalence of self-reported known hypertension of participants in three studies was 13% [20], 7% [38], and 6% [57]. One study reported the prevalence of objective hypertension for females as 14% [32].

HDL cholesterol and triglycerides were reported in four studies [3,27,33,57] (*n* = 1640), revealing pooled means of 1.3 ± 0.9 mmol/L and 19 ± 1.6 mmol/L, respectively. Additionally, three studies reported the prevalence of low HDL as 8% [21], 46% [27], and 57% [26] and of elevated triglycerides as 24% [21], 28% [27], and 29% [26]. The pooled mean of three studies [3,33,57] (*n* = 1337) reporting TC was 5.3 ± 1.0 mmol/L, and an LDL cholesterol mean of 3.3 ± 0.9 mmol/L was derived from two studies [3,33] (*n* = 742). The prevalence of self-reported known dyslipidemia of participants in two studies was 10% [57] and 19% [38]. Only two studies investigated hyperglycemia, reporting the prevalence as 31% [21] (≥100 mg/dL) and 30% [27] (≥5.6 mmol/L). 

### 3.5. Behavioral Cardiometabolic Risk Factors among Pilots

Thirty-one studies included the evaluation of behavioral cardiometabolic risk factors. Alcohol intake was investigated in 10 samples of airline pilots [6,24,27,33,36,38,39,46,47,57,60,61]; one study utilized a validated questionnaire [36], and five studies [27,33,36,38,39] (*n* = 2538) ascertained “regular alcohol intake” on the basis of a participant self-recall question, producing a pooled prevalence of 52% (51.3–53.1). Twelve studies [6,26,27,29,32,33,36,37,39,49,57,60] (*n* = 19,116) reported smoking prevalence, yet no studies evaluated quantity or frequency of smoking. The pooled prevalence was 9.4% (9.3–9.5%). One study reported the prevalence of smoking for females as 6% [32].

From the 20 studies evaluating sleep, seven studies objectively measured sleep hours with actigraphy (*n* = 1764) [29,35,48,50,51,52,53,59], and six used self-recall methods (*n* = 2224) [17,24,39,56,59,62]. The pooled means for objective and self-recall sleep hours per night were 7.2 ± 1.1 and 7.0 ± 0.6, respectively. Three studies reported the prevalence of <6 h of sleep per night as 23% [43], 20% [61], and 22% [20]. Furthermore, other studies reported that <6 h of sleep per night was associated with obesity [41] and poor sleep quality [42] within participants. The prevalence of excessive sleepiness assessed by the Epworth Sleepiness Scale (score ≥ 10) was reported by five studies [19,20,39,42,46], exhibiting a pooled prevalence of 44.5% (44.1–44.8%). Among four studies reporting high obstructive sleep apnea (OSA) risk ascertained from the Berlin Questionnaire, the prevalence was 5% [19], 20% [39], 21% [42], and 29% [20], providing a pooled mean of 21.4% (21.3–21.5%).

The prevalence of self-reported insufficient physical activity (<150 min MVPA per week) was reported in five studies (*n* = 2233) providing a pooled prevalence of 51.5% (51.3–51.7%) [22,26,42,56,62]. Additionally, <150 min MVPA per week was found to be associated with obesity in one study which reported a prevalence ratio of 1.08 (0.98–1.19) [41]. One study reported the mean days per week of moderate physical activity and strenuous physical activity as 3.3 ± 1.9 and 2.0 ± 1.4, respectively. Another study reported the mean walking minutes and MVPA minutes per week as 110 ± 117 and 145 ± 72, respectively.

Three studies (*n* = 955) reported the prevalence of subjective insufficient daily fruit intake as 33% (<200 g/day) [38], 60% [56], and 65% (<2 servings/day) [62] and of insufficient daily vegetable intake as 19% (<300 g/day) [38], 47% [62], and 48% [56] (<3 servings/day). From these studies, two reported the prevalence of combined insufficient fruit and vegetable intake as 68% [56] and 84% [62]. One study reported the mean number of snacks per duty as 4 ± 3 [60].

### 3.6. Psychological Cardiometabolic Risk Factors among Pilots

Sixteen studies included an evaluation of psychological cardiometabolic risk factors. Among 10 studies investigating the prevalence of psychological fatigue, four studies (*n* = 2987) utilized the Fatigue Severity Scale (FSS), two of which reported a psychological fatigue prevalence (FSS ≥ 4 mean score) of 77% [54] and 89% [46,47]. Another two studies reported the severe psychological fatigue prevalence (FSS ≥ 36 total score) as 33% [19] and 68% [20]. The prevalence of elevated psychological fatigue in the remaining studies (*n* = 2719) was reported as 5% [58], 27% [42,43], 30% [60,61], and 75% [34], each produced with different methodology.

Seven studies subjectively measured the prevalence of depression, with a pooled mean of 21% (20.8–21.6) for mild depression derived from five studies [19,25,30,54,58] (*n* = 3411) utilizing the Patient Health Questionnaire (PHQ-9; score ≥ 10). One study reported a depression prevalence of 35% [20] according to the Hospital Anxiety and Depression Scale (score > 8), whereas another study reported depression or anxiety within the last 12 months as 54.4% [40]. One study reported mild depression (PHQ-9 score ≥ 10) prevalence in females as 11% [58]. Two studies (*n* = 2527) reported the prevalence of mild anxiety derived from the Generalized Anxiety Disorder-7 (GAD-7; score > 10) scale, noting 4% [30] and 7% [54]. The prevalence of below-average or poor subjective self-rated health was reported in three studies (*n* = 1282) as 8% [38], 25% [56], and 39% [22], each derived from different methodology.

## 4. Discussion

To our knowledge, this is the first comprehensive synthesis of published research pertaining to physiological, behavioral, and psychological cardiometabolic health risk factors among this unique occupational group. Our findings provide stakeholders including aviation medical professionals, policymakers, researchers, clinicians, and occupational health authorities with a scientific synthesis of the magnitude of prevalence of cardiometabolic health risk factors among commercial pilots. These findings may be beneficial to inform developments in aviation health practices and policies to support pilot health and wellness, to mitigate risks of occupational morbidity, medical conditions causing loss of license, and medical incapacity, and to support flight safety [5].

Findings from the review suggest similar health risk factor prevalence to the general population, yet higher sleep duration, less physical activity, lower smoking rates, higher regular alcohol consumption, less obesity, and a higher overall rate of body mass index >25 among pilots. We discovered, within the literature reviewed, a dominance of self-reported data, with most studies exhibiting moderate to high risk of methodological bias. Indeed, there are limited high-quality studies within the field, warranting the need for future research to address the gaps within and strengthen the body of knowledge.

### 4.1. Prevalence of Physiological Cardiometabolic Risk Factors among Pilots

As described by the International Civil Aviation Organization (ICAO) aviation medical regulations, cardiometabolic health risk data are acquired routinely during aviation medical examinations for pilots >35 years old for CVD risk assessment, which include BMI, BP, resting heart rate, blood lipids, and HbA1c [5]. In 2015, the global prevalence of overweight, obesity, and overweight plus obesity in the general population was reported as 38.7%, 16.4%, and 55.1%, respectively [63]. This general population estimate is relative to the country, age, and sex characteristics represented in the present review of studies conducted among pilots. Past research has reported a lower prevalence of overweight and obesity in pilots compared to the general population [6,64,65]. Indeed, the present review found that pilots had a 4.7% lower prevalence of obesity than the general population [63]. As obesity is a major risk factor for diseases such as CVD and T2D [66], the lower rate of obesity within pilots may promote a lower pilot population cardiometabolic disease relative risk compared to the general population. 

Interestingly, with overweight and obesity pooled together, we discovered that pilots had an overall 3.5% higher rate of overweight plus obesity compared to the general population (58.6% and 55.1%, respectively). This finding suggests that past reports of lower rates of overweight and obesity within pilots [6,64,65] compared to the general population may be archaic, and future research should investigate the underlying causal mechanisms that contribute to overweight and obesity rates among pilots. A noteworthy consideration for interpretation of this information is the lack of random sampling and potential response bias within studies on pilots compared to the general population, which adds a notable limitation to the validity of prevalence comparisons between populations.

Most countries represented within the present review were high-income countries. In 2010, the global hypertension prevalence was estimated as 31.1% (30.0–32.2%), while that among high-income countries was estimated as 28.5% (27.3–29.7%) [67]. We found four studies reporting the prevalence of hypertension within pilots, ranging from 11% to 29% [23,27,32,56], with a pooled prevalence slightly lower than the general population at 27.6%. Airline pilots undergo regular medical examinations evaluating BP [5], and this regular active monitoring may promote the observed lower hypertension prevalence. However, one study reported a 38% [21] prevalence of pilots exhibiting elevated BP (≥130/85 mmHg); thus, it would be valuable for future epidemiological research to report the prevalence of elevated BP, accompanying hypertension rates in order to better inform researchers on the distribution of BP ranges across the pilot population.

Prospective epidemiological studies have consistently reported that unfavorable blood lipid profiles are associated with increased incidence of metabolic syndrome (MS) and NCDs such as CVD [68]. We found three studies reporting the prevalence of markers of dyslipidemia in pilots, with prevalence of low HDL ranging from 8% to 57% and of elevated TG ranging from 24% to 29% [21,26,27], whereas MS prevalence ranged from 15% [21] to 38% [27]. These evident variances in prevalence rates based on the small sample of studies (*n* = 3) [21,26,27] illustrate the need for further research to be conducted regarding quantification of these risk factors in pilots to reach valid inferences of their prevalence.

The global prevalence of diabetes is rising, and it has been estimated that 49.7% of people living with diabetes are undiagnosed [69]. Investigation of impaired glucose tolerance and hyperglycemia is scarce in the airline pilot literature, with only two studies reporting the prevalence of hyperglycemia (30.4–31.3%) [21,27] and scant research reporting on the prevalence of elevated HbA1c, which is the leading diagnostic criteria for T2D [69]. This dearth of information may be attributable to past barriers for diabetic pilots to operate commercials flights due to the risk of incapacitation from hypoglycemia while flying, yet recent advances in insulin therapies, monitoring techniques, and modes of administration have given rise to policy developments reducing barriers for diabetic pilots to operate commercial flights [70]. Seemingly, there is a need for more research attention on glycemic control and identifying the prevalence of elevated risk markers for T2D among airline pilots.

### 4.2. Prevalence of Behavioral Cardiometabolic Risk Factors among Pilots

From the 31 studies we found reporting on behavioral risk factors, sleep was the most frequently reported risk factor (*n* = 21). Sleep disruption is an inherent risk for pilots as occupational characteristics such as extended duty periods, work schedules, crossing time zones, and sleep restrictions cause perturbance of sleep routine consistency [71]. Recent research has indicated that sleep difficulty is frequently expressed as a primary source of work induced stress among pilots [25]. The present review found the mean pooled sleep hours per night to range from 7.0 h to 7.1 h, indicating that the population mean falls within the lower range of sleep guidelines for health in adults, which has been reported as the attainment of 7–9 h [72]. We found three studies reporting the prevalence of short sleep (<6 h) ranging from 20% to 23% [20,59,61], which is comparably lower than a USA-based study noting a prevalence of ≤6 h sleep as 29% among the general population in 2012 [73]. Indeed, due to the influence of fatigue on flight safety, pilots are often subject to fatigue management training via aviation medical management [71,74], facilitating the implementation of adaptive coping strategies to mitigate fatigue which may support the attainment of sleep guidelines within the population.

Past research has reported lower sex- and age-adjusted prevalence for smoking and higher levels of physical exercise among aircrews compared to the general population [65]. Indeed, we found a pooled smoking prevalence of 9% among pilots, which is considerably lower than a 2015 prevalence estimate of 25% for smoking among the global male general population [75]. Interestingly, for physical activity, we found a pooled prevalence of 51.5% for insufficient physical activity among pilots, which is markedly higher than a recent global prevalence estimate of 32% (30–33%) in 2016 among the general population within high-income countries [76]. However, our findings were only derived from four studies using self-recall data and small samples, making comparisons with the general population of scant validity. Future research utilizing objective outcome measures is important to further evaluate the accuracy of current findings. 

Alcohol use is a leading risk factor for global disease burden, with the global prevalence of current drinking estimated as 47% in 2017 [77]. According to five studies we found, the pooled mean for regular alcohol intake among pilots was 52%. However, the lack of quantity and the low-quality methodology among studies minimize the validity of prevalence estimation. Furthermore, pilots may be inherently biased to misrepresent their true alcohol intake to aviation medical professionals or researchers due to aviation regulations prohibiting alcohol consumption within 8 h of acting as a crew member and existing alcohol testing mandates [78].

Dietary behaviors are a leading risk factor for obesity and cardiometabolic diseases such as CVD and T2D [79]. Previous studies have conveyed occupational factors such as inconsistent mealtimes, physical inactivity on duty, suboptimal airport and airline catering options, and shift work as factors that may be detrimental to healthy dietary patterns among pilots [6,7,21]. There is a dearth of literature pertaining to the quantification of dietary behaviors among pilots, with only two studies identified in this review reporting the prevalence of pilots who were not achieving daily fruit and vegetable intake guidelines of ≥5 servings ranging from 68–84% [56,62]. Although lacking validity, this estimate is comparable to the estimated global prevalence estimate of 79% derived from the World Health Survey 2002–2004 [80], relative to the country and sex characteristics represented in the present review.

### 4.3. Prevalence of Psychological Cardiometabolic Risk Factors among Pilots

High levels of psychological fatigue are associated with elevated risk of CVD and excess mortality within the general population [11] and are detrimental to a pilot’s ability to safely operate the aircraft or perform safety-related duties [5]. We found the prevalence of elevated psychological fatigue to range from 5% to 77% [19,42,43,46,47,54,58,60,61] And of severe psychological fatigue to range from 33% to 68% [19,20]. The heterogeneity of methodology among studies within pilots inhibits valid comparisons with the general population. Nonetheless, with numerous studies reporting noteworthy rates of elevated psychological fatigue during nonduty periods, this warrants further research regarding the development of innovative interventions to better facilitate fatigue mitigation in this occupational group.

Major depressive disorder is associated with elevated cardiometabolic risk factors and poor health outcomes [12]. Depression and mental health issues among pilots have been proposed as contributing factors in numerous flight incidents resulting in mass casualties [40]. Thus, psychological risk factors are pertinent to pilot health and wellness and, in turn, flight operation safety. We discovered a pooled mean of 21% for mild depression [19,25,30,54,58] among pilots. Comparatively, a prevalence of 21% for mild depression was reported within a general population sample using congruent methodology, delineating a similar prevalence between populations.

### 4.4. Study Strengths

To the authors’ knowledge there is no published scientific synthesis of cardiometabolic health risk factor data for airline pilots. The studies included in the systematic review were derived from 20 different countries from around the globe. Therefore, the findings are not localized to a certain region and are relevant data pertaining to the global airline pilot population. This review revealed insights that diverge from previous assumptions regarding cardiometabolic health among airline pilots, thus providing useful data which may inform public health practice and the development of targeted initiatives to support occupational health and safety.

### 4.5. Study Limitations

As this review sought to identify baseline nonwork duty-related prevalence of cardiometabolic health risk factors, we did not examine risk factor quantification during or immediately following flight duty periods, as these work characteristics often elicit acute inflated risk prevalence for factors such as psychological fatigue, sleep disruption, and other psychological distress-related parameters [39,52]. Thus, the present review did not capture the magnitude of work duty-induced perturbations to behavioral and psychological cardiometabolic health risks.

Furthermore, as we sought to identify the prevalence of cardiometabolic health risks among the overall airline pilot population, we did not stratify outcomes by fleet division, such as short-haul, long-haul, or mixed-fleet. The comparison of health risk prevalence between fleet divisions may be an appropriate scope for a future systematic review, which would add to the literature for understanding the magnitude of health risk difference between pilot rosters. 

Due to the heterogeneity of publication dates among the literature featured in our review, the global general population prevalence comparison studies utilized may not optimally align with timepoints from studies among pilots and should be considered by readers with our presented population comparisons. Additionally, the heterogeneity of measurements of cardiometabolic parameters among the airline pilot studies reviewed and the general population estimates should be considered in the interpretation of our findings. 

Lastly, the low quantity of robust studies limits the generalizability of the current findings reported within the literature. Future high-quality epidemiological research utilizing validated measurements will be valuable to increase the probability of attaining reliable conclusions pertaining to the health risk prevalence within the pilot population. To provide further meaningful insight into pilot cardiometabolic health risk and to address gaps in the literature, research attention pertaining to the assessment of glycemic control (i.e., HbA1c) and blood lipids, objectively measured health behaviors (dietary behaviors, physical activity, alcohol intake), and wider assessment of depressive symptoms among the airline pilot population would provide valuable contributions to advance the body of knowledge.

## 5. Conclusions

The findings of this review provide synthesis on the prevalence and magnitude of cardiometabolic health risk factors among airline pilots. A wide range of prevalence rates were reported for many investigated health risk parameters in the literature, with pervasiveness of overweight and obesity, insufficient physical activity, elevated psychological fatigue, insufficient fruit and vegetable intake, and regular alcohol consumption among pilots. The inherent bias, dominance of self-report data, and heterogeneity of methodology mean that it was not possible to establish strong conclusions. Future research utilizing objective measures and robust random sampling strategies are advocated to strengthen the validity of prevalence estimates and enhance the generalizability of findings.

## Figures and Tables

**Figure 1 ijerph-19-04848-f001:**
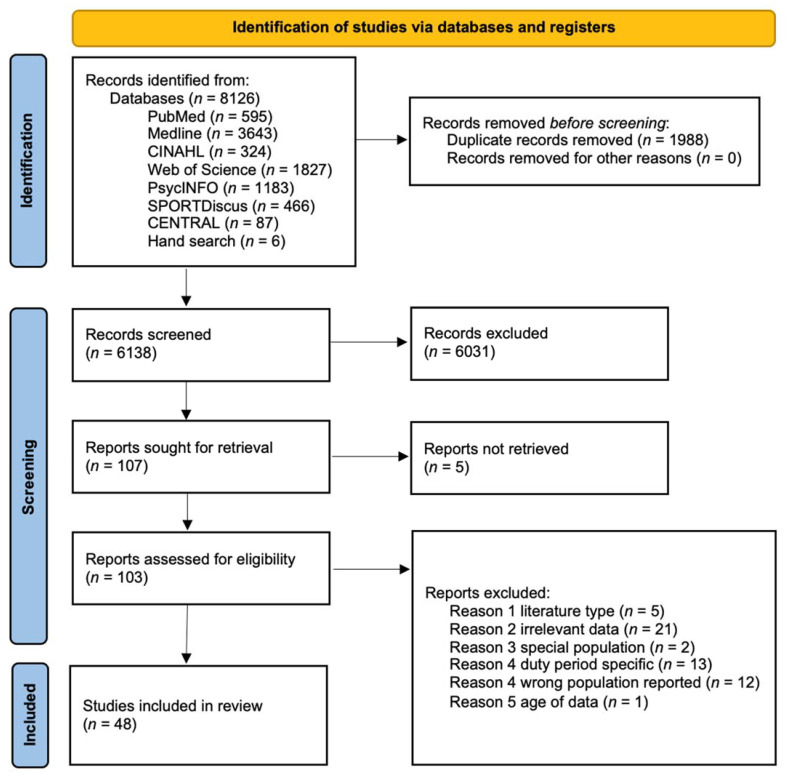
PRISMA flow diagram. PRISMA, Preferred Reporting Items for Systematic Reviews and Meta-Analyses.

**Figure 2 ijerph-19-04848-f002:**
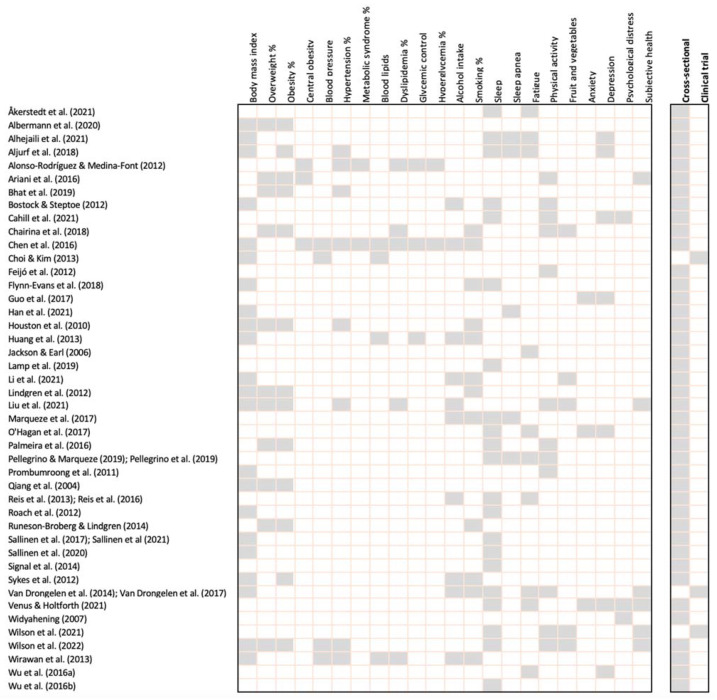
Cardiometabolic risk markers and airline pilot outcome summary for each study.

**Table 1 ijerph-19-04848-t001:** Search terms blocks were combined for text and word search in PubMed and adapted to the remaining databases: 1 and 2; 1 and 3; 1, 2, and 3.

1. Airline Pilots	2. Cardiometabolic Risk Markers	3. MeSH
Pilots OR “airline pilot *” OR “commercial pilot *” OR “professional pilot *” OR “civil pilot *” OR “civilian pilot *” OR “aviation pilot *” OR “commercial airline *” OR aircrew OR “cockpit crew *” NOT military * NOT army NOT “pilot study” NOT piloted NOT “pilot project” NOT “pilot research”	“Health risk *” OR “risk factor *” OR cardiometabolic OR cardio-metabolic OR cardiovascular OR “cardiometabolic risk” OR “metabolic syndrome” OR “syndrome x” OR diabetes OR hypertension OR weight OR overweight OR obesity OR “body composition” OR adiposity OR “physical activity” OR exercise OR sleep OR circadian OR apnoea OR apnea OR nutrition OR diet OR eating OR fruit * OR vegetable * OR stress OR lipids OR cholesterol OR glucose OR insulin OR “insulin resistance” OR “insulin sensitivity” OR “waist circumference” OR fat OR “blood pressure” OR hypertension OR “C-reactive protein” OR “inflammatory markers” OR inflammation OR “microvascular dysfunction” OR fatigue OR medical OR depression OR stress OR distress OR anxiety OR alcohol OR smok * OR microalbumin * OR “endothelial dysfunction”	MeSH terms: “risk factors” [mesh] OR “health risk behaviors” [mesh] OR “health status indicators” [mesh] OR “risk assessment” [mesh]

Note: * indicates use of truncation.

**Table 2 ijerph-19-04848-t002:** Methodological quality scores of cross-sectional studies.

Author (Year)	External Validity				Internal Validity						
	1	2	3	4	5	6	7	8	9	10	Quality
Åkerstedt et al. (2021) [17]	N	N	N	Y	Y	Y	Y	Y	Y	Y	(3) High
Albermann et al. (2020) [18]	Y	Y	N	Y	Y	Y	N	Y	Y	Y	(2) High
Alhejaili et al. (2021) [19]	N	N	N	N	Y	Y	Y	Y	Y	Y	(4) Med
Aljurf et al. (2018) [20]	Y	N	N	N	Y	Y	Y	Y	Y	Y	(3) High
Alonso-Rodríguez and Medina-Font (2012) [21]	Y	Y	N	Y	Y	Y	Y	N	Y	Y	(2) High
Ariani et al. (2017) [22]	N	N	N	N	N	Y	N	Y	Y	Y	(6) Med
Bhat et al. (2019) [23]	Y	Y	N	N	Y	Y	Y	Y	Y	Y	(2) High
Bostock and Steptoe (2012) [24]	Y	N	N	N	Y	Y	N	Y	Y	Y	(4) Med
Cahill et al. (2021) [25]	N	N	N	N	Y	Y	Y	Y	N	Y	(5) Med
Chairina et al. (2018) [26]	N	N	N	N	Y	N	N	N	Y	Y	(7) Low
Chen et al. (2016) [27]	Y	N	N	N	Y	Y	Y	Y	N	Y	(4) Med
Feijó et al. (2012) [28]	Y	Y	N	N	Y	Y	Y	Y	Y	Y	(2) High
Flynn-Evans et al. (2018) [29]	N	N	N	Y	Y	Y	N	Y	Y	Y	(4) Med
Guo et al. (2017) [30]	Y	N	N	N	Y	Y	Y	Y	N	Y	(4) Med
Han et al. (2020) [31]	N	N	N	N	Y	Y	Y	Y	N	Y	(5) Med
Houston et al. (2010) [32]	Y	Y	Y	Y	Y	Y	Y	Y	Y	N	(1) High
Huang et al. (2012) [33]	N	Y	N	Y	Y	Y	N	N	N	Y	(5) Med
Jackson and Earl (2006) [34]	N	N	N	N	Y	Y	N	Y	N	Y	(6) Med
Lamp et al. (2019) [35]	N	N	N	N	Y	Y	Y	Y	N	Y	(5) Med
Li et al. (2021) [36]	N	N	N	N	Y	Y	Y	Y	Y	Y	(4) Med
Lindgren et al. (2012) [37]	Y	Y	N	N	Y	Y	N	Y	N	Y	(4) Med
Liu et al. (2021) [38]	N	Y	N	Y	Y	Y	N	Y	N	Y	(4) Med
Marqueze et al. (2017) [39]	Y	Y	N	N	Y	Y	Y	Y	Y	Y	(2) High
O’Hagen et al. (2016) [40]	Y	N	N	N	Y	Y	N	Y	Y	Y	(4) Med
Palmeira et al. (2016) [41]	Y	Y	Y	N	Y	Y	N	Y	Y	Y	(2) High
Pellegrino and Marqueze (2018) [42]	N	N	N	N	Y	Y	Y	Y	Y	Y	(4) Med
Pellegrino et al. (2018) [43]	N	N	N	N	Y	Y	Y	Y	Y	Y	(4) Med
Prombumroong et al. (2011) [44]	N	N	N	N	Y	Y	Y	Y	Y	Y	(4) Med
Qiang et al. (2004) [45]	N	Y	N	Y	Y	Y	Y	Y	Y	Y	(2) High
Reis et al. (2013) [46]; Reis et al. (2016) [47]	Y	N	N	N	Y	Y	N	Y	Y	Y	(4) Med
Roach et al. (2012) [48]	N	N	N	N	Y	Y	Y	Y	N	Y	(5) Med
Runeson-Broberg and Lindgren (2013) [49]	Y	Y	N	N	Y	Y	N	Y	N	Y	(4) Med
Sallinen et al. (2017) [50]	Y	N	Y	Y	Y	N	Y	N	N	N	(5) Med
Sallinen et al. (2020) [51]	Y	N	N	Y	Y	N	Y	N	Y	N	(5) Med
Sallinen et al. (2021) [52]	N	N	N	N	Y	Y	N	Y	Y	Y	(5) Med
Signal et al. (2014) [53]	N	N	N	N	Y	Y	Y	Y	Y	Y	(4) Med
Sykes et al. (2012) [6]	Y	Y	N	N	Y	Y	N	Y	Y	Y	(3) High
Venus and Holtforth (2021) [54]	N	N	N	N	Y	Y	Y	Y	Y	Y	(4) Med
Widyahening (2007) [55]	N	N	N	N	Y	N	N	Y	N	Y	(7) Low
Wilson et al. (2022) [56]	Y	Y	Y	N	Y	Y	Y	Y	Y	Y	(1) High
Wirawan et al. (2013) [57]	Y	Y	N	N	Y	Y	N	N	N	Y	(5) Med
Wu et al. (2016a) [58]	N	N	N	N	Y	Y	Y	Y	Y	Y	(4) Med
Wu et al. (2016b) [59]	Y	N	N	N	Y	Y	Y	Y	Y	Y	(3) High

Note: High = high quality (low risk of bias); Low = low quality (high risk of bias); Med = medium quality (moderate risk of bias); N, no; Y, yes; 1—Was the study’s target population a close representation of the national population in relation to relevant variables, age, sex, and occupation? 2—Was the sampling frame a true or close representation of the target population? 3—Was some form of random selection used to select the sample OR was a census undertaken? 4—Was the likelihood of nonresponse bias minimal? 5—Were data collected directly from the subjects (as opposed to a proxy)? 6—Was an acceptable case definition used in the study? 7—Was the study instrument that measured the parameter of interest (e.g., prevalence of lower-back pain) shown to have reliability and validity (if necessary)? 8—Was the same mode of data collection used for all subjects? 9—Was the length of the shortest prevalence period for the parameter of interest appropriate? 10—Were the numerator(s) and denominator(s) for the parameter of interest appropriate?

**Table 3 ijerph-19-04848-t003:** Risk-of-bias assessment of clinical trials.

Author (Year)	1	2	3	4	5	6	7
Choi and Kim 2013 [3]	High	High	High	High	Unclear	Low	Low
Van Drongelen et al. 2014 [60]; Van Drongelen et al. 2016 [61]	Low	High	High	High	Low	Low	High
Wilson et al. 2021 [62]	High	High	High	Low	Low	Low	Low

Note: 1 = random sequence; 2 = allocation concealment; 3 = blinding of participants; 4 = blinding of outcomes; 5 = incomplete outcome data; 6 = selective reporting; 7 = other; High = high risk of bias; Low = low risk of bias; Unclear = not possible to rate risk of bias.

## Data Availability

Not applicable.

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
