# Peer review of "The Prevalence of Cardiometabolic Health Risk Factors among Airline Pilots: A Systematic Review"

_ijerph, 2022, doi:10.3390/ijerph19084848_

Round 1
Reviewer 1 Report
The study is interesting, but major changes must be made, and the modifications I have made be taken into account, so that it can be published

Author Response
Comments and Suggestions for Authors: The study is interesting, but major changes must be made, and the modifications I have made be taken into account, so that it can be published
Author Response: Thank you to reviewer 1 for taking the time to review our manuscript. The edits and suggestions provided below have improved the overall quality of the paper. We have made some amendments based on your suggestions, and our individual responses to your comments are listed below.
Q1: An electronic search of PubMed, 16 MEDLINE (via OvidSP)
It is more correct to say An electronic search was conducted utilizing MEDLINE (via OvidSP), and ( via Pubmed)
A1: Thank you for this suggestion, this correction has been completed.
Q2: Search terms blocks were combined for text and word search in PubMed, and adapted to remaining databases: 1 77 and 2; 1 and 3; 1 and 2 and 3.
But they do not explain the descriptors used in the different databases consulted, as well as the search strategies
MeSH terms to consider
- Occupational Health*
A2: Thank you for your MeSH term suggestion, we will consider this term for future reviews, however for the present review we perceive the utilized terms adequate to extract the relevant data for this manuscript.
Q3: literature published between January 1, 1990, and February 1, 2022.
I think the dates are wrong. Well I guess it would be until February 2021 and not 2022 as the manuscript says??
A3: Thanks for your comment, the year 2022 is indeed correct, however we have updated the day to February 28 instead of February 1.
Q4: Bibliographic to consider
A4: Thank you for your suggestions, the study by Wilson et al 2022 that has recently been published has been integrated into the review now. Regarding the other suggestions, one of which was already included in the review, and the two others were already initially screened during the literature search but did not satisfy the inclusion criteria for this review.
Q5: Recommendations to follow: This study is interesting, but it has methodological limitations, such as evaluating prevalence in studies that are clinical trials, which measure incidence rather than prevalence.
On the other hand, this work has not been able to carry out a meta-analysis, and they do not make it clear why it has not been done... and it is that it presents a high heterogeneity. In addition, objective results are analyzed and compared with subjective results (self-administered surveys).
In order to be published, my recommendation is to eliminate the clinical trial studies. And leave only in cross-sectional or prevalence studies and in any case include observational studies of cases and controls.
Systematic reviews (SRs) of prevalence a study are increasingly necessary, but until now no fully structured and accepted methodology has been proposed to ensure high quality procedure. The most useful tools to measure the quality in prevalence studies is Joanna Briggs Institute in Australia after standardization and measurement of reliability among observer , and Agency for Healthcare Research and Quality, el Scottish Intercollegiate Guidelines Group
A5: Thank you for your insights and comments. Regarding your recommendation to eliminate the clinical trial studies, we deemed this appropriate to include as these studies reported on risk factors that have been scarcely reported among other literature. Given the limited quantity of published studies pertaining to these risk factors among airline pilots, we perceive these studies of value to include in this review. The present review intends to synthesize the global literature available pertaining to cardiometabolic health risk factors among airline pilots to reveal to the research community what currently exists, so that future research may build upon the existing body of knowledge. Since these clinical trials report valuable information for the purpose of this review, and indeed similar systematic review methodology has been implemented in other published systematic reviews such as the reference below, we deem them appropriate to remain in the manuscript.
https://onlinelibrary.wiley.com/doi/full/10.1002/1348-9585.12150
Nonetheless, if the reviewer further insists we remove the clinical trials, we are happy to discuss this further.
Reviewer 2 Report
Wilson et al in this review looked at the prevalence of cardiometabolic risk factors among airline pilots. This is an important evaluation since this information may be relied upon in preventive strategies for this population.
Below are some comments and suggestions based on my review;
Abstract
Line 22 – 24: Consider organizing the risk factors here according to whether they increase or decrease risk.
Materials and Methods
Line 122: consider writing the author name (for example “adapted from Hoy et al”) rather than saying “adapted from 13”
Lines 175 – 180: Consider revising this section. See if it's more appropriate to either use letters or numbers to list the number of studies in each country. Also the phrase “…four involved participants from numerous countries” seems out of place (may be better at the end of the section).
Results
Line 210: Is the term “physical cardiometabolic risk factors” standard. I found it confusing at first.
Line 233 and 241: Are BP and HDL considered physical CM risk?
Line 233: Consider adding BP to parathesis “….hypertension (BP >= 140/90)”
Line 242: Consider adding US units (mg/dL) depending on the readership of the journal to enhance understanding of findings.
Line 253: Consider defining “regular alcohol” to help with understanding (number or frequency of drinking)
Line 269: PA is the same abbreviation for Pennsylvania in line 112. Consider using unique abbreviations.
Line 281: Is psychological cardiometabolic risk factor a standard term. Is fatigue a psychological or physical condition?
Discussion
Line 313: Consider if “…self-reported…” is more appropriate.
Line 315: Consider if “…body of knowledge” is more appropriate
Line 347: Having 2 different definitions of BP 140/90 vs 130/85 may confuse the interpretation of the findings.
Line 402: Consider changing to “…inherently biased…”
Author Response
Comments and Suggestions for Authors: Wilson et al in this review looked at the prevalence of cardiometabolic risk factors among airline pilots. This is an important evaluation since this information may be relied upon in preventive strategies for this population.
Below are some comments and suggestions based on my review;
Author Response: Thank you to reviewer 2 for taking the time to review our manuscript. The edits and suggestions provided below have improved the overall quality of the paper. We have made amendments based on your suggestions, and our individual responses to each of your comments are listed below.
ABSTRACT
Q1: Line 22 – 24: Consider organizing the risk factors here according to whether they increase or decrease risk.
A1: Thank you for this suggestion, this correction has been completed.
MATERIALS AND METHODS
Q2: Line 122: consider writing the author name (for example “adapted from Hoy et al”) rather than saying “adapted from 13”
A2: Thank you for this suggestion, this correction has been completed.
Q3: Lines 175 – 180: Consider revising this section. See if it's more appropriate to either use letters or numbers to list the number of studies in each country. Also the phrase “…four involved participants from numerous countries” seems out of place (may be better at the end of the section).
A3: Thank you for this suggestion, the correction to use numbers for each country and relocation of the indicated sentence to the end has been completed.
RESULTS
Q4: Line 210: Is the term “physical cardiometabolic risk factors” standard. I found it confusing at first.
A4: Thanks for pointing out that this may be a source of confusion for the reader. We have changed this terminology throughout to “physiological” health risk factor to enhance clarity.
Q5: Line 233 and 241: Are BP and HDL considered physical CM risk?
A5: Physiological cardiometabolic health risk factors are those that relate to physical physiological markers that are established as risk factors (i.e. not subjectively reported risk factors such as behavioral and psychological risk factors). Thus, BP, blood lipids, along with other metabolic syndrome related risk factors such as BMI, central adiposity, and glycemic control are deemed as “physiological” risks under our term “physiological” risk factors which we perceive to be aligned with the literature base.
Q6: Line 233: Consider adding BP to parathesis “….hypertension (BP >= 140/90)”
A6: Thank you for this suggestion, this correction has been completed.
Q7: Line 242: Consider adding US units (mg/dL) depending on the readership of the journal to enhance understanding of findings.
A7: Thank you for this suggestion, we are happy to make this amendment if the journal requires this terminology to be used.
Q8: Line 253: Consider defining “regular alcohol” to help with understanding (number or frequency of drinking)
A8: Thanks for your comment, as mentioned on line 253 only one of the studies found utilized a validated questionnaire and the other studies did not provide a definition in their methodology of what constituted “regular alcohol” intake. Thus, for us to provide a definition of regular alcohol intake may not accurately represent the data. Thus, we have not included a definition and have noted throughout the manuscript the evident heterogeneity in methods should be considered by the reader in the interpretation of our findings.
Q9: Line 269: PA is the same abbreviation for Pennsylvania in line 112. Consider using unique abbreviations.
A9: Thank you for pointing this out, we have removed this abbreviation.
Q10: Line 281: Is psychological cardiometabolic risk factor a standard term. Is fatigue a psychological or physical condition?
A10: Psychological cardiometabolic health risk factors as a term is largely neglected in the current literature base. However, as cited in our introduction, numerous psychological health risk factors that we have explored in this review are indeed independently associated with cardiometabolic health outcomes and therefore are necessary to examine in this review.
Fatigue is poorly defined and understood in the literature, and it is associated with both physical physiological and psychological aspects. Physical fatigue (i.e. fatigue from physical exertion) and psychological fatigue (i.e. self-reporting fatigue level) are both often referred to in the literature as ‘fatigue’. We have included fatigue as a ‘psychological’ health risk factor within this study as the nature of quantification of fatigue from the literature pertaining to perceived fatigue among pilots has been via self-report subjective instruments that quantify an individual subjective fatigue level. Throughout the manuscript we have changed the terminology of “fatigue” to “psychological fatigue” to enhance clarity for the reader.
DISCUSSION
Q11: Line 313: Consider if “…self-reported…” is more appropriate.
A11: Thank you for this suggestion, this correction has been completed.
Q12: Line 315: Consider if “…body of knowledge” is more appropriate
A12: Thank you for this suggestion, this correction has been completed.
Q13: Line 347: Having 2 different definitions of BP 140/90 vs 130/85 may confuse the interpretation of the findings.
A13: Thank you for pointing this out, however in this case on line 347 130/85 was referred to as elevated blood pressure, which is indeed a different threshold to hypertension and congruent with other literature. Thus, we view it appropriate to remain as described, however we are happy to discuss further if the reviewer insists on specific amendments to this.
Q14: Line 402: Consider changing to “…inherently biased…”
A14: Thank you for this suggestion, this correction has been completed.
Reviewer 3 Report
Thank you for the invite to review this manuscript. This systematic review explores the prevalence of cardiometabolic risk factors in airline pilots. The review highlights this is an important population to consider given their occupational demands. The review provides oversight into the prevalence of various risk factors and highlights future research that should be conducted to develop knowledge in this area further.
The manuscript would be strengthened if the following comments were addressed:
Introduction
General: The background and rationale for the study would be strengthened if the authors could include some details/data regarding factors such as the number of airline pilots. It would also be beneficial to provide some more detailed context regarding the occupational demands/typical work demands individuals working as pilots experience, for example, typical hours sleep, break between scheduled flights etc.
Line 39: For clarity it should be stated that the sample of airline pilots were only from Korea, otherwise it could be interpreted that this represents global data
Lines 40-41: For clarity it should be stated that the sample of airline pilots were only from the UK, otherwise it could be interpreted that this represents global data
Line 45: Can the authors cite any more recent publications that support the occupational demands of pilots. Some of the cited references are from 2012, therefore such data may now no reflect work demands. For example, this study present more recent data of sleepiness and fatigue: https://doi.org/10.1016/j.ssci.2020.104833
Line 53: Please revise ‘pertaining to evaluation of…’ to ‘pertaining to the evaluation of’
Line 55: What is meant by ‘appropriate allocation of resources’? Is this referring to interventions to reduce risk factors in pilots, or treatments for pilots. Please provide further details around this point to make this clear and strengthen the justification of the systematic review
Method
Line 70: What was the rationale for start the search from 1990? Please provide justification of this
Line 96: Please remove additional spaces in the line
Line 100: Please define VO2, as this is the first time it has been used in the text
Lines 120-133: Details around the quality assessment process adopted are a little unclear. It is described that clinical trials were evaluated using a tool from Cochrane. Was this the Newcastle-Ottawa scale? Two different types of classification are described based on scoring or quality, which creates some confusion.
Results
Figure 1: Asterisks are included in the figure which are not defined in the figure caption beneath. Also, can the authors be more specific with their reasons for excluding studies after the full text screening phase. For example, 14 records were excluded due to the wrong population, which seems large considering the initial search criteria should have ensured a specific population were captured, in addition to the title and abstract screening.
Lines 211-216: Please include measurement units for BMI data
Lines 233-239: For the paragraph reporting the blood pressure data, please include the sample sizes
Lines 276-280: For the paragraph reporting the diet data, please include the sample sizes
Lines 282-287: For the paragraph reporting the psychological data, please include the sample sizes
Discussion
General: The authors have compared the data to the general population, however in these samples, do the authors know the occupations of the included sample? Could it be that some of the general population in these samples are employed in similar high stress/shift work etc. jobs. This should be considered.
General: Have the authors considered how this prevalence data compares to any similar occupational data
Lines 309-311: The authors have compared the findings from their review to the general population, yet no references are provided for where this general population data derives from.
Lines 331-333: When the authors compare the overweight/obesity data form pilot to the general population, have these data been collected using the same approach (e.g., self-report, physician assessed). This should be considered in this discussion point.
Limitations: The authors should comment on the heterogeneity of measurements used to assess different cardiometabolic parameters. They should also highlight any outcomes that were not captures from the review that future research should explore to provide further insight into pilot’s cardiometabolic health risk
Appendix A
Please define abbreviations used in the footnote of the table
Please ensure data is reported to a consistent number of decimal places
Author Response
Comments and Suggestions for Authors: Thank you for the invite to review this manuscript. This systematic review explores the prevalence of cardiometabolic risk factors in airline pilots. The review highlights this is an important population to consider given their occupational demands. The review provides oversight into the prevalence of various risk factors and highlights future research that should be conducted to develop knowledge in this area further.
The manuscript would be strengthened if the following comments were addressed:
Author Response: Thank you so much to reviewer 3 for taking the time to review our manuscript. The edits and suggestions provided below have improved the overall quality of the paper. We have made amendments based on your suggestions, and our individual responses to each of your comments are listed below.
INTRODUCTION
Q1: General: The background and rationale for the study would be strengthened if the authors could include some details/data regarding factors such as the number of airline pilots. It would also be beneficial to provide some more detailed context regarding the occupational demands/typical work demands individuals working as pilots experience, for example, typical hours sleep, break between scheduled flights etc.
A1: Thank you for your comment, this is appreciated. Occupational characteristics of airline pilots in general are congruent with information provided on lines 42-45, yet specific details on hours of sleep, breaks between scheduled flights etc. vary depending on the type of roster a pilot is undertaking, such as whether they are a long-haul or short-haul pilot. Since the focus of the present review was airline pilots overall as discussed on lines 452-456, we think additional description of work demands are irrelevant to this body of work. Further, this review aimed to quantify the pooled results of measures such as sleep hours across the population, which has not been previously reported.
In regard to describing the global number of pilots, this has not been reported in peer reviewed journal articles, which were the criteria for information inclusion in this study, thus there is not an established accurate evidence-based number to provide. Further, the COVID-19 pandemic has had a profound effect on layoffs globally of airline pilots, further making accurate quantification of current global pilot numbers uncertain.
Nevertheless, if the reviewer insists on other specific information to be provided, we are happy to discuss this further.
Q2: Line 39: For clarity it should be stated that the sample of airline pilots were only from Korea, otherwise it could be interpreted that this represents global data
A2: Thank you for this suggestion, this correction has been completed.
Q3: Lines 40-41: For clarity it should be stated that the sample of airline pilots were only from the UK, otherwise it could be interpreted that this represents global data
A3: Thank you for this suggestion, this correction has been completed.
Q4: Line 45: Can the authors cite any more recent publications that support the occupational demands of pilots. Some of the cited references are from 2012, therefore such data may now no reflect work demands. For example, this study present more recent data of sleepiness and fatigue: https://doi.org/10.1016/j.ssci.2020.104833
A4: Thank you for this suggestion, this correction has been completed.
Q5: Line 53: Please revise ‘pertaining to evaluation of…’ to ‘pertaining to the evaluation of’
A5: Thank you for this suggestion, this correction has been completed.
Q6: Line 55: What is meant by ‘appropriate allocation of resources’? Is this referring to interventions to reduce risk factors in pilots, or treatments for pilots. Please provide further details around this point to make this clear and strengthen the justification of the systematic review
A6: Thanks for your suggestion, the sentence has been amended to the following: “Estimations of health risk prevalence are important for monitoring of trends and to inform risk reduction interventions, hence the aim of the current review is to critically analyse the global literature to quantify the prevalence of modifiable cardiometabolic health risk factors among commercial airline pilots”.
METHODS
Q8: Line 70: What was the rationale for start the search from 1990? Please provide justification of this
A8: Our date range of from 2022 back to 1990 was similar to numerous other NCD prevalence systematic reviews. Further, studies report notable changes in global prevalence of various NCDs and associated major risk factors within the last decade compared to 1980. Thus, like other research it was deemed appropriate to limit our time frame to the past few decades as earlier prevalence estimates may be less representative of current trends. Supportive reference links to the preceding statements are provided below:
https://www.ncbi.nlm.nih.gov/pmc/articles/PMC6076564/
https://www.ncbi.nlm.nih.gov/pmc/articles/PMC3295618/
https://www.ncbi.nlm.nih.gov/pmc/articles/PMC5780690/
https://www.thelancet.com/journals/lancet/article/PIIS0140-6736(16)00618-8/fulltext
https://academic.oup.com/eurpub/article/29/Supplement_4/ckz185.196/5624434
Q9: Line 96: Please remove additional spaces in the line
A9: Thank you for this suggestion, this correction has been completed.
Q10: Line 100: Please define VO2, as this is the first time it has been used in the text
A10: Thank you for this suggestion, this correction has been completed.
Q11: Lines 120-133: Details around the quality assessment process adopted are a little unclear. It is described that clinical trials were evaluated using a tool from Cochrane. Was this the Newcastle-Ottawa scale? Two different types of classification are described based on scoring or quality, which creates some confusion.
A11: Thank you for pointing out the typo in this section referring to the Newcastle-Ottawa scale, we have amended the wording to refer to the cross-sectional study assessment instead.
RESULTS
Q12: Figure 1: Asterisks are included in the figure which are not defined in the figure caption beneath. Also, can the authors be more specific with their reasons for excluding studies after the full text screening phase. For example, 14 records were excluded due to the wrong population, which seems large considering the initial search criteria should have ensured a specific population were captured, in addition to the title and abstract screening.
A12: Thanks for pointing out the asterisk typo, this has been amended. Regarding the 14 studies that were excluded due to the wrong population, these were cases where pilots were reported in the title/abstract, however upon analysis of the study methodology it was revealed the study population did not satisfy our inclusion criteria. Examples of the wrong population were Air Force or combat pilots, rotary wing pilots, airline pilot sub-populations e.g. overweight or obese.
Q13: Lines 211-216: Please include measurement units for BMI data
A13: Thank you for this suggestion, this correction has been completed.
Q14: Lines 233-239: For the paragraph reporting the blood pressure data, please include the sample sizes
A14: Thank you for this suggestion, this correction has been completed.
Q15: Lines 276-280: For the paragraph reporting the diet data, please include the sample sizes
A15: Thank you for this suggestion, this correction has been completed.
Q16: Lines 282-287: For the paragraph reporting the psychological data, please include the sample sizes
A16: Thank you for this suggestion, this correction has been completed.
DISCUSSION
Q17: General: The authors have compared the data to the general population, however in these samples, do the authors know the occupations of the included sample? Could it be that some of the general population in these samples are employed in similar high stress/shift work etc. jobs. This should be considered.
A17: Thanks for your insight which is appreciated. The general population data utilized to make comparisons with findings on pilots are indeed “general population” based, thus general population data does not represent a specific occupation but the global population at large. As stated in the discussion we utilized global general population data available that was relative to the countries represented by literature on airline pilots included in the present review.
Q18: General: Have the authors considered how this prevalence data compares to any similar occupational data
A18: Thanks for your comment, the purpose of this review was to identify the prevalence of cardiometabolic health risk factors in airline pilots primarily and we made comparisons to global general population data as a logical comparison in our discussion. Comparative analysis between airline pilots and other occupational groups is outside of the scope/purpose of the present review.
Q19: Lines 309-311: The authors have compared the findings from their review to the general population, yet no references are provided for where this general population data derives from.
A19: Thank you for your comment on this, the indicated sentence is an introductory summative statement of findings overall, and all general population data utilized for comparison to health risk factor findings among pilots are included subsequently throughout the discussion section.
Q20: Lines 331-333: When the authors compare the overweight/obesity data form pilot to the general population, have these data been collected using the same approach (e.g., self-report, physician assessed). This should be considered in this discussion point.
A20: Thank you for your comment on this, the general population data literature are based on large scale international reviews of studies reporting the health risk factor in question among adults globally. These reviews include a combination of studies that have used self-report and objectively measured outcome measures. Indeed, this was the case among the global airline pilot literature that we reviewed also. To address your comment, we have added mention of this factor to the limitations section as follows: “Due to the heterogeneity of publication dates among literature featured in our review, the global general population prevalence comparison studies utilized may not optimally align with timepoints from studies among pilots and should be considered by readers with our presented population comparisons. Additionally, the heterogeneity of measurements of cardiometabolic parameters among the airline pilot studies reviewed and the general population estimates should be considered in interpretation of our findings. Finally, the lacking quantity of robust studies limits the generalizability of the current findings reported within the literature. Future high quality epidemiological research will be valuable to increase the probability of attaining reliable conclusions pertaining to the health risk prevalence within the pilot population.”
Q21: Limitations: The authors should comment on the heterogeneity of measurements used to assess different cardiometabolic parameters. They should also highlight any outcomes that were not captures from the review that future research should explore to provide further insight into pilot’s cardiometabolic health risk
A21: Thank you for this comment, regarding the heterogeneity of measurements, detail has been added as described in the previous question response. To address your comment regarding future research, the following additional detail has been added to the end of the limitations section: “To provide further meaningful insight into pilot’s cardiometabolic health risk and to address gaps in the literature, research attention pertaining to assessment of glycemic control (i.e. HbA1c) and blood lipids, objectively measured health behaviors (dietary behaviors, physical activity, alcohol intake), and wider assessment of depressive symptoms among the airline pilot population would provide valuable contributions to advance the body of knowledge”.
Q22: Appendix A
- Please define abbreviations used in the footnote of the table
- Please ensure data is reported to a consistent number of decimal places
A22: Thank you for these suggestions, these corrections have been completed.
Round 2
Reviewer 1 Report
Recommendations
See paragraph 70: A broad search strategy was implemented to gather literature published published between January 1, 1990, and February 28, 2022.
Q1: You have to check the dates of the searches. It is better to search for complete years, that is why I indicated in the previous review, that the search is more appropriate until 12-31-2021
A1: Thanks for your comments, we appreciate your advice. Nevertheless, we conducted the search according to the dates described in the manuscript that are in alignment with our PROSPERO review registration, we intend for the search dates to remain as described. Indeed, other studies have implemented search strategies that end within a year as we did, for example: https://pubmed.ncbi.nlm.nih.gov/22521520/
Q2:
Say that they have not modified what I told you, the database is Medline, but it has 2 forms of entry, one is Pubmed and the other is by Ovid.
The correct thing to say is that the Medline database was consulted through (Pubmed and Ovid) PubMed is a search system in the Internet that provides free access to MEDLINE, the main database Biomedical at the National Library of Medicine of the United States (NLM, for its acronym in English)
A2: We reported MEDLINE (Via OvidSP) as this was the method used to search MEDLINE. Other studies have reported similarly, for example https://pubmed.ncbi.nlm.nih.gov/22521520/ https://onlinelibrary.wiley.com/doi/full/10.1002/1348-9585.12150
Q3:
At the level of statistical analysis, only a description is made and nothing else is explained, it would be convenient to explain the issue of heterogeneity as it was calculated and the publication bias that has not been explained either.
It would be interesting to explain the following:
1.-A sensitivity analysis has not been performed for each risk factor or grouped by physical activity, obesity, cardiovascular risk factors. The degree of heterogeneity has not been taken into account or calculated.
2.-The risk of bias was assessed for each study. We separately calculated the prevalence of. Subgroup analysis and meta-regression analyses were performed to determine the sources of heterogeneity. We performed Cochran's Q test and the I2 test statistic to assess the heterogeneity.
3.-Publication bias was evaluated using Begg's test and Egger's test.?? Publication bias has not been calculated.
It would be of great interest if you provided this information because it is a limitation of the study as it has a high degree of heterogeneity that the authors comment on but do not evaluate.
A3: Due to the heterogeneity of methods among the literature, a narrative analysis was conducted and a random or fixed effects meta-analysis was not conducted due to the lacking quantity among the literature. Thus, weighted pooled mean estimates were performed. Accordingly, we discussed the presence of heterogeneity among the literature throughout our manuscript to emphasize this to the reader as a limitation of current findings. Due to the limited number of included articles, publication bias was not assessed.
Various other prevalence related systematic reviews with similar literature characteristics and methodology to the present review have not included sensitivity analyses with assessment of prevalence with inherent heterogeneity and lacking quantity of studies among the literature, some examples listed below:
https://onlinelibrary.wiley.com/doi/full/10.1002/1348-9585.12278
https://onlinelibrary.wiley.com/doi/full/10.1002/1348-9585.12150
https://www.tandfonline.com/doi/full/10.1080/14789949.2020.1859588